# Impact of COVID-19 on out-of-hospital cardiac arrest: A registry-based cohort-study from the German Resuscitation Registry

Patrick Ristau[1], Jan Wnent[1,2,3], Jan-Thorsten Gräsner[1,2], Matthias Fischer[4], Andreas Bohn[5,6], Berthold Bein[7], Sigrid Brenner[8], Stephan Seewald[1,2]*

1 Institute for Emergency Medicine, University Hospital Schleswig-Holstein, Kiel, Germany, 2 Department of Anesthesiology and Intensive Care Medicine, University Hospital Schleswig-Holstein, Campus Kiel, Kiel, Germany, 3 School of Medicine, University of Namibia, Windhoek, Namibia, 4 Department of Anesthesiology, Intensive Care Medicine and Emergency Medicine, ALB-FILS Kliniken, Göppingen, Germany, 5 Fire Department, City of Münster, Münster, Germany, 6 Department of Anaesthesiology, Intensive Care and Pain Medicine, University Hospital Münster, Münster, Germany, 7 Department of Anaesthesiology, Intensive Care Medicine, Emergency Medicine and Pain Therapy, ASKLEPIOS Klinik St. Georg, Hamburg, Germany, 8 Department of Anaesthesiology and Intensive Care Medicine, University Hospital Dresden, Dresden, Germany

* stephan.seewald@uksh.de

**Data Availability Statement:** All relevant data are within the manuscript and its Supporting Information files (S1 and S2 Tables).

## Abstract

### Introduction

The global COVID-19 pandemic effects people and the health system. Some international studies reported an increasing number of out-of-hospital cardiac arrest (OHCA). Comparable studies regarding the impact of COVID-19 on incidence and outcome of OHCA are not yet available for Germany.

### Materials and methods

This epidemiological study from the German Resuscitation Registry (GRR) compared a non-pandemic period (01.03.2018–28.02.2019) and a pandemic period (01.03.2020–28.02.2021) regarding the pandemic-related impact on OHCA care.

### Results

A total of 18,799 cases were included. The incidence of OHCA (non-pandemic 117.9 vs. pandemic period 128.0/100,000 inhabitants) and of OHCA with resuscitation attempted increased (66.0 vs. 69.1/100,000). OHCA occurred predominantly and more often at home (62.8% vs. 66.5%, p<0.001). The first ECG rhythm was less often shockable (22.2% vs. 20.3%, p = 0.03). Fewer cases of OHCA were observed (58.6% vs. 55.6% p = 0.02). Both the bystander resuscitation rate and the proportion of telephone guided CPR remained stable (38.6% vs. 39.8%, p = 0.23; and 22.3% vs. 22.5%, p = 0.77). EMS arrival times increased (08:39 min vs. 09:08 min, p<0.001). Fewer patients reached a return of spontaneous circulation (ROSC) (45.4% vs. 40.9%, p<0.001), were admitted to hospital (50.2% vs. 45.0%, p<0.001), and discharged alive (13.9% vs. 10.2%, p<0.001).

**Funding:** The authors received no specific funding for this work.

**Competing interests:** I have read the journal's policy and the authors of this manuscript have the following competing interests: PR works as the scientific coordinator of the German Resuscitation Registry. JW, JTG, MF, AB, SB, and SS are members of the steering committee of the German Resuscitation Registry. JTG is the spokesperson of the steering committee.

## Discussion

Survival after OHCA significantly decreased while the bystander resuscitation rate remained stable. However, longer EMS arrival times and fewer cases of witnessed OHCA may have contributed to poorer survival. Any change to EMS systems in the care of OHCA should be critically evaluated as it may mean a real loss of life—regardless of the pandemic situation.

## Introduction

Since beginning of 2020, health systems worldwide have been confronted with the pandemic occurrence of the SARS-CoV-2 pathogen and the COVID-19 illnesses caused by it. The direct death rates are well documented: 376.2 million confirmed cases including 5.7 million deaths have been reported to WHO [1]. Reliable data are also available for Germany: since the beginning of the pandemic, 10.2 million people have been diagnosed with the virus [2]. By February 2022, at least 118,200 people in Germany had died from COVID-19 [2].

However, the global pandemic not only has direct effects on people. Due to the systemic effect that epidemics have on other parts of the health system, there may also be effects on the care of patients without a recent infectious episode [3]. For example in Germany, there are reports of decreasing numbers of patients seeking medical help during the Covid-19 pandemic [4, 5]. Furthermore, this also seems to have impact on the outcome of time-critical emergencies, e.g. more arrhythmia and higher troponin levels after ST-Elevation Myocardial Infarction (STEMI) have been shown during the lockdown caused by Covid-19 [6].

Comparable studies regarding the impact of COVID-19 on out-of-hospital cardiac arrests (OHCA) are not yet available for Germany. At the same time, there are some international reports of increased numbers of OHCA [7, 8]. The care for people with OHCA may provide as surrogate parameter for assessing the performance of an Emergency Medical Service (EMS), as OHCA is a frequent emergency with an incidence of 135.4 cases/100,000 inhabitants per year [9, 10]. In Germany, resuscitation is attempted in slightly more than half of the cases, with an incidence of 72.6 resuscitations/100,000 inhabitants per year [9].

The COVID pandemic has led to structural changes in the EMS work on scene: Ambulance dispatchers query a possible corona infection before sending EMS personnel [11]; bystanders alerted by an app or first responders are not alerted or more cautiously alerted [12]; and the putting on of personal protective equipment presumably prolongs the arrival time at the patient's location [13, 14].

The interplay of these modifications is assumed to have various effects on system and process quality as well as outcome. That is why this study analysed the changes in incidence, quality of care and outcome of patients with out-of-hospital cardiac arrest (OHCA) in the context of the COVID 19 pandemic in comparison with a pre-pandemic period.

## Materials and methods

### Study design and participants

The present study is an epidemiological cross-sectional study as defined by a population-based cohort study. It is reported according to the STrengthening the Reporting of OBservational studies in Epidemiology (STROBE) statement [15].

The German Resuscitation Registry (GRR) has been previously described [16, 17]. Since its foundation in 2007, up to now 135 emergency medical services, more than 210 in-hospital

medical emergency and resuscitation teams and over 70 Cardiac Arrest Centres from Germany, Austria and Switzerland have joined. More than 350,000 anonymised records of primary, secondary, and long-term care have been collected so far and are available as a database. The GRR offers participants an online benchmarking function and has a comprehensive reporting system. In addition, it carries out public relations work and publishes cross-regional quality and annual reports [18].

Participation in the German Resuscitation Registry is voluntary. The participating EMS enter all OHCA cases regardless of resuscitation was attempted or not. Some EMS demonstrate particularly high data quality (high data quality group), defined as follows[9]:

- Incidence of resuscitation > 30 cases/100,000 inhabitants per year,

- ROSC (Return of Spontaneous Circulation) rate < 80%,

- RACA (ROSC after Cardiac Arrest) score [19] calculable > 60%, and

- Proportion of documented in-hospital follow-up data of at least 30%.

To compare the pandemic-related impact on OHCA care with care prior to COVID-19, two comparative year periods were set and compared:

- Non-pandemic period (01.03.2018–28.02.2019)

- Pandemic period (01.03.2020–28.02.2021)

Furthermore, each of these annual periods was divided into quarters (Q):

- Q1 (March to May)

- Q2 (June to August)

- Q3 (September to November)

- Q4 (December to February)

To increase the validity of the presented study, only EMS that showed particularly high data quality in these two years were included. In this way, it was possible to look at and analyse seasonal deviations in addition to year-on-year comparisons.

## In- and exclusion of cases

All cases of OHCA from the two comparison periods in EMS with high data quality in both periods were included.

## Study protocol and ethical clearance

A priori, a study protocol was written and approved by the scientific advisory board of the German resuscitation registry under the processing number 2021–05. The study retrospectively analysed completely anonymised data from the German Resuscitation Register. Therefore, the consent of the participants was not required. This procedure was approved by the Ethics Committee of the Medical Faculty of the Kiel University (CAU) under reference number D 460/21.

## Statistical analysis

The statistical analyses were carried out with SPSS 28 (IBM). Here, in addition to descriptive parameters such as mean and standard deviation (SD), p-values were calculated where meaningful by applying Pearson's Chi-Square-Test, two-sided T-Test following Levene's test or Mann-Whitney-U-Test, whereby a p-value < .05 was considered significant.

## Results

During the observation periods, a total of 54,440 cases of Out-of-Hospital Cardiac Arrest (OHCA) were recorded in the online database of the GRR. Of these, after excluding 33,961 cases not in the high data quality group, and 1,680 from sites not in both years in the high data quality group, a total of 18,799 cases originated from 25 high data quality sites were included, representing a population of 7,599,142 inhabitants in 2018/2019 and 7,683,759 inhabitants in 2020/2021. These cases consisted of 10,324 OHCA with resuscitation attempted (non-pandemic period: 5,016, pandemic period: 5,308) and 8,475 OHCA without resuscitation attempted (non-pandemic-period: 3,946, pandemic period: 4,529) (Fig 1). Based on the 10,324 cases in which resuscitation was attempted, the more extensive evaluations presented in Table 1 were calculated.

The incidence of all OHCA increased relevantly in the pandemic-period (117.9 vs. 128.0/ 100,000 inhabitants). The proportion of cases in which no resuscitation attempts were made, for example, due to DNAR orders, futility, or when reliable signs of death were present, increased significantly (44.0% vs. 46.0%, p = 0.006). Finally, the incidence of OHCA with resuscitation attempted increased between the two comparison periods (66.0/100,000 vs. 69.1/ 100,000).

Comparing 5,016 cases with resuscitation attempted in the non-pandemic period (01.02.2018–28.02.2019) with 5,308 cases in the pandemic period (01.03.2020–28.02.2021) yielded no difference in the sex and age distribution of the patients with OHCA. There are predominantly men in both groups (65.2% vs. 66.0% p = 0.38), with an average age of just under 70 years (69.7 vs. 69.7 years, p = 0.53). Likewise, the suspected cause of OHCA did not differ between groups with predominantly a presumed cardiac cause (73.0% vs. 73.1%, p = 0.97), followed by asphyxia (15.0% vs. 14.5%, p = 0.52) (Table 1).

Fewer cases of OHCA were observed (58.6% vs. 55.6% p<0.001), and both the bystander resuscitation rate and the proportion of telephone guided CPR remained stable (38.6% vs. 39.8%, p = 0.23; and 22.3% vs. 22.5%, p = 0.77, respectively). Alerted first responders initiated resuscitation in slightly fewer cases during the pandemic period (4.3% vs. 4.2%, p = 0.72).

At the same time, the first-arriving EMS vehicle needed significantly more time to reach the patient (8:39 min vs. 09:08 min, p<0.001) (Fig 2) and a smaller proportion of patients were reached in less than eight minutes (59.3% vs. 54.0%, p<0.001).

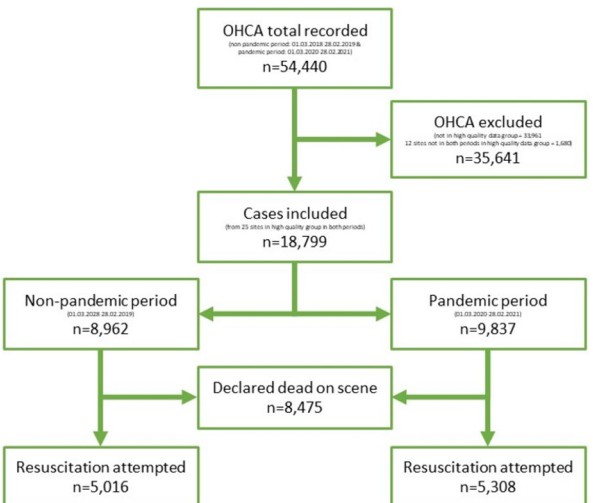

**Fig 1. Flowchart of in- and exclusion.**

**Table 1. Baseline characteristics, initial management, and outcome of OHCA.**

| | Non-pandemic period (01.03.2018–28.02.2019) | Pandemic period (01.03.2020–28.02.2021) | Missing | p-value |
|---|---|---|---|---|
| High-quality data sites | 25 | 25 | n/a | |
| Inhabitants covered | 7,599,142 | 7,683,759 | n/a | |
| Total OHCA in high-quality data sites (with/without resuscitation attempts) | 8,962 | 9,837 | n/a | < .001 |
| • Incidence of OHCA (cases/inhabitants) | 117.9/100,000 | 128.0/100,000 | | |
| OHCA without resuscitation attempts | 3,946 | 4,529 | n/a | < .001 |
| • Incidence of OHCA without resuscitation attempts (cases/inhabitants) | 51.9/100,000 | 58.9/100,000 | | |
| • Proportion of cases without resuscitation attempts | 44.0% | 46.0% | | .006 |
| OHCA with resuscitation attempts | 5,016 | 5,308 | n/a | .02 |
| • Incidence of OHCA with resuscitation attempts (cases/inhabitants) | 66.0/100,000 | 69.1/100,000 | | |
| Sex (males/females) | 65.2%/34.8% | 66.0%/34.0%% | 0 | .38 |
| Age (years) | 69.7 (SD 16.9) | 69.7 (SD 16.6) | 78 | .53 |
| Cause of cardiac arrest | | | 0 | .81 |
| • Cardiac cause and missing | 3,663 (73.0%) | 3,878 (73.1%) | | |
| • Trauma | 127 (2.5%) | 153 (2.9%) | | |
| • Asphyxia | 750 (15.0%) | 770 (14.5%) | | |
| • Intoxication | 69 (1.4%) | 76 (1.4%) | | |
| • Others | 407 (8.1%) | 431 (8.1%) | | |
| Location of cardiac arrest | | | 21 | .003 |
| • At home | 3,145 (62.8%) | 3,519 (66.5%) | | |
| • Workplace | 101 (2.0%) | 104 (2.0%) | | |
| • In public | 791 (15.8%) | 772 (14.6%) | | |
| • Other locations | 973 (19.4%) | 898 (17.0%) | | |
| Collapse witnessed | 2,941 (58.6%) | 2,952 (55.6%) | 0 | .02 |
| • Layperson | 2,189 (43.6%) | 2,200 (41.4%) | | |
| • First Responder | 76 (1.5%) | 80 (1.5%) | | |
| • EMS | 676 (13.5%) | 672 (12.7%) | | |
| • Not witnessed | 2,075 (41.4%) | 2,356 (44,4%) | | |
| Bystander CPR | 1,938 (38.6%) | 2,112 (39.8%) | 0 | .23 |
| Telephone CPR | 1,118 (22.3%) | 1,196 (22.5%) | 0 | .77 |
| First responder CPR | 217 (4.3%) | 222 (4.2%) | | .72 |
| Arrival of first EMS vehicle within 8 minutes | 2,895 (59.3%) | 2,757 (54.0%) | 334 | < .001 |
| Median response time, call to arrival of first EMS vehicle (minutes) | 08:39 (SD 04:19) | 09:08 (SD 04:40) | 334 | < .001 |
| Shockable rhythm at EMS arrival | 1,111 (22.2%) | 1,073 (20.3%) | 31 | .02 |
| Tracheal intubation | 3,301 (65.8%) | 3,327 (62.7%) | 0 | < .001 |
| Mechanical CPR | 666 (13.3%) | 718 (13.5%) | 0 | .71 |
| ROSC at any time | 2,277 (45.4%) | 2,167 (40.9%) | 6 | < .001 |
| • Incidence of ROSC at any time (cases/inhabitants) | 30.0/100,000 | 28.2/100,000 | | |
| Transport to hospital (minutes) | 13:59 (SD 09:45) | 13:58 (SD 09:32) | 5,856 | .89 |
| Hospital admission | 2,516 (50.2%) | 2,391 (45.0%) | 1 | < .001 |
| • Hospital admission with ROSC | 1,890 (37.7%) | 1,736 (32.7%) | | |
| • Hospital admission with ongoing CPR | 626 (12.5%) | 655 (12.3%) | | |
| • Declared dead on scene | 2,499 (49.8%) | 2,917 (55.0%) | | |
| Cases with available hospital data | 2,289 (91.0%) | 1,880 (78.6%) | | < .001 |
| Survival at hospital discharge | 645 (13.9%) | 475 (10.2%) | 1,054 | < .001 |
| • Incidence of survival at hospital discharge (cases/inhabitants) | 8.5/100.000 | 6.2/100.000 | | |

CPR = cardiopulmonary resuscitation; EMS = emergency medical service; OHCA = out-of-hospital cardiac arrest; ROSC = return of spontaneous circulation

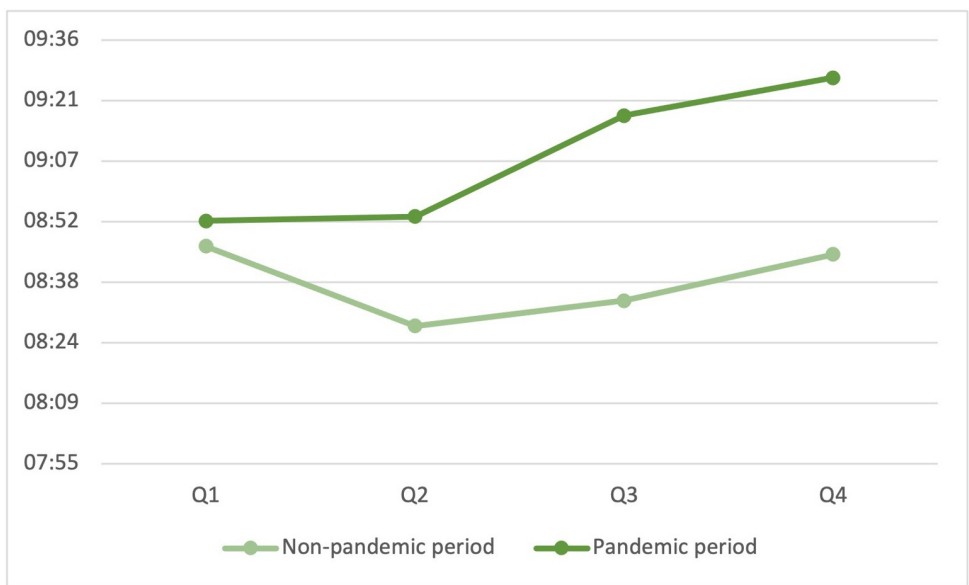

**Fig 2. Median response time.**

There was a difference regarding the places where the OHCA occurred: While the overall number of OHCAs that occurred at workplace remained constant (2.0% vs. 2.0%, p = 0.85), comparatively less occurred in public (15.8% vs. 14.6%, p = 0.09), and more occurred at home (62.8% vs. 66.5%, p<0.001).

In fewer cases, the first derived ECG rhythm showed a shockable rhythm (22.2% vs. 20.3%, p = 0.02). Patients were less often endotracheally intubated (67.8% vs. 62.7%, p = <0.001). The proportion of cases in which mechanical CPR (mCPR) was performed did not change significantly (13.3% vs. 13.5%, p = 0.71).

Nevertheless, significantly fewer patients reached a return of spontaneous circulation (ROSC) (44.6% vs. 40.9%, p<0.001). Even though transport times to hospital remained approximately the same (13:59 min vs. 13:58 min, p = 0.89), fewer patients were admitted to hospital (45.4% vs. 40.9%, p<0.001), including a slightly decreased proportion being admitted under ongoing CPR (12.5% vs. 12.3%, p = 0.83). More patients were declared dead on scene (48.8% vs. 55.0%, p<0.001).

Overall, significantly fewer patients were discharged alive from hospital after out-of-hospital cardiac arrest (13.9% vs. 10.2%, p<0.001).

This impression is emphasized by a closer look at the different quarters of the two comparison periods: In quarters 2 to 4, the arrival times become disproportionately longer during the pandemic phase (08:28 to 08:45 min vs. 08:54 to 09:27 min) (Fig 2), and overall survival decreased reciprocally (16.8 to 12.6% vs. 12.2 to 6.1%) (Fig 3).

## Discussion

This study shows the impact of a pandemic on the health care system exemplified by the out-of-hospital care of patients with cardiac arrest in Germany. During the pandemic period March 2020 to February 2021 the survival after OHCA significantly decreased while the proportion of cases without resuscitation attempts increased significantly. This is remarkable, as the patient population that suffered an OHCA did not change—in particular, the vast majority of patients are assumed to have suffered OHCA due to a cardiac cause -, and the bystander

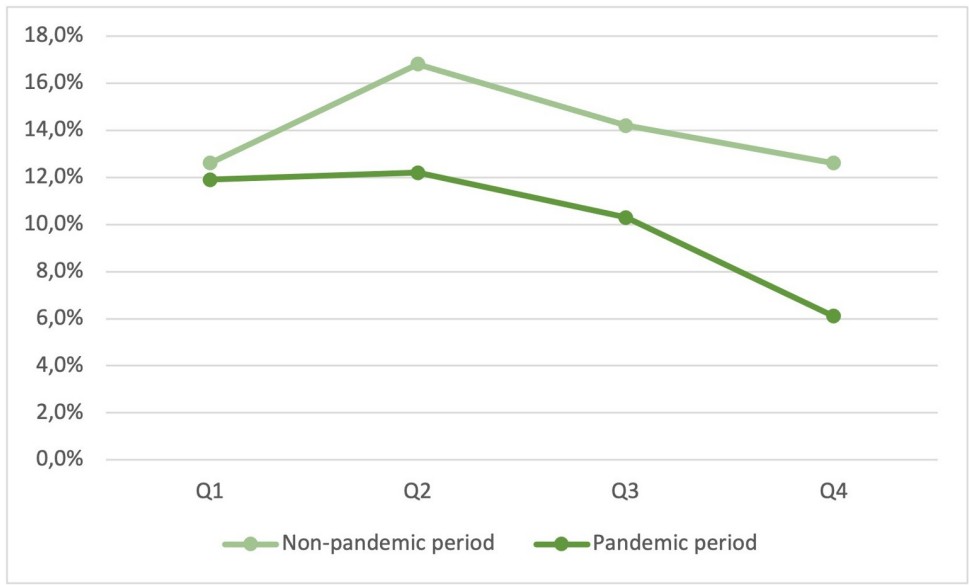

**Fig 3. Survival at hospital discharge.**

resuscitation remained constant. But longer arrival times of the EMS and fewer witnessed OHCA contributed to poorer survival in out-of-hospital cardiac arrests. Similar observations have been reported previously, particularly concerning the increasing incidence of OHCA, the decreasing proportion of initially schockable rhythms and poorer survival [7, 20–23].

These results should also be seen in the context of the recent COVID-19-related excess mortality of 4.9% in Germany for the year 2021 [24]. This does not even include those deceased that could have been saved but died, for example, due to a prolonged arrival time in the case of cardiac arrest. The decreased number of patients who were discharged alive from hospital after OHCA equates to a relative decrease in survival of nearly 23% when comparing OHCA survival during the pandemic period with the previous non-pandemic period, or an additional 3 out of 100 people with OHCA who might have been potentially rescued under different circumstances.

The previously reported decrease in bystander CPR [25] could not be confirmed in this study. It is reasonable to assume that due to the temporary shutdown and the increased use of home-office options and the resulting shift in emergency locations from the public to the home, a consistent number of bystanders—now no longer in the form of colleagues and passers-by, but instead through relatives—nevertheless provided first aid. In fact, it is reasonable to assume that when the reduced proportion of witnessed OHCA is combined with a constant proportion of bystander and telephone-guided CPR, this results in an inferiority baseline situation for the patient. That is also shown by the decrease in shockable rhythms on arrival of the ambulance service.

Our study cannot make any definite statement about why longer response times occurred. This may be explained with longer intervals in the dispatch centre due to questions about the infection status of the patient or because of more time needed to establish the full personal protective equipment before starting the EMS car or getting to the patient. In addition, it was described for many local authorities that first responder systems were no longer alerted during the COVID 19 pandemic and have also not been put back into operation to date. Corresponding differences during the pandemic in resuscitation practices and reduced access to guideline-recommended care have been reported previously [20, 23].

The in-hospital post-resuscitation care was not investigated by our study. Therefore, the influence of intensive care units that were otherwise busy with the care of COVID-19 patients on the outcome cannot be assessed. However, it can be assumed that the reduced survival was primarily due to poorer out-of-hospital variables, as the proportion of patients who were admitted to hospital with ROSC was significantly worse, despite a comparable patient population.

Whether mortality correlates with the level of COVID-19 incidence remains an open question [26]. In addition, we cannot comment on differences between COVID-19-positive and COVID-19-negative patients in this study because no information is available on the individual infection status of the included patients. However, a worse outcome of COVID-19-positive patients is known from the literature [8]. In addition, the limitations that apply to retrospective studies must be observed [27].

The care of people with cardiac arrest can be seen as a quality indicator for the overall performance of the emergency medical service. In view of our results, it can be assumed that the overall performance measured by patient outcome of the German emergency medical services deteriorated during the pandemic and in particular over the course of time. This also resulted in fewer endotracheal intubations during pandemic period, which cannot be explained otherwise but is even more surprising as a protected airway reduces the risk of infection für EMS staff. Whether the COVID-19 pandemic-related negative effects described here also apply to other medical conditions needs to be clarified in further studies.

## Limitations

The present study is a retrospective study. Due to missing outcome data, some data sets had to be excluded from further analysis. Due to the organizational structures of the emergency medical services included, the transferability of this study to other systems may be limited.

## Conclusion

For future pandemics, it must be critically questioned and included in disaster preparation that every change in an EMS system in the care of out-of-hospital cardiac arrest can mean a real loss of human life—regardless of the pandemic situation. Accordingly, pandemic events must be included in process design for out-of-hospital emergency care [20].

## Supporting information

**S1 Table. Dataset definition of minimal underlying data.**
(XLSX)

**S2 Table. Minimal underlying dataset.**
(XLSX)

## Acknowledgments

We thank all emergency medical technicians, paramedics and physicians participating in the German Resuscitation Registry for the certainly difficult data entry and updating of the database in times of the COVID-19 pandemic.

## Author Contributions

**Conceptualization:** Patrick Ristau, Jan Wnent, Jan-Thorsten Gräsner, Stephan Seewald.

**Formal analysis:** Patrick Ristau, Stephan Seewald.

**Resources:** Patrick Ristau, Jan-Thorsten Gräsner.

**Supervision:** Jan Wnent.

**Visualization:** Patrick Ristau, Stephan Seewald.

**Writing – original draft:** Patrick Ristau.

**Writing – review & editing:** Jan Wnent, Jan-Thorsten Gräsner, Matthias Fischer, Andreas Bohn, Berthold Bein, Sigrid Brenner, Stephan Seewald.

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
