## [Decision Letter · Decision Letter 0]

6 Jun 2022

PONE-D-22-03386Impact of COVID-19 on Out-of-Hospital Cardiac Arrest: A Registry-Based Cohort-Study from the German Resuscitation RegistryPLOS ONE

Dear Dr. Seewald,

Thank you for submitting your manuscript to PLOS ONE. After careful consideration, we feel that it has merit but does not fully meet PLOS ONE’s publication criteria as it currently stands. Therefore, we invite you to submit a revised version of the manuscript that addresses the points raised during the review process.

We look forward to receiving your revised manuscript.

Kind regards,

Andrea Ballotta

Academic Editor

PLOS ONE

**Journal requirements:**

2. In the ethics statement in the Methods and online submission information, please ensure that you have specified what type you obtained (for instance, written or verbal, and if verbal, how it was documented and witnessed). If your study included minors, state whether you obtained consent from parents or guardians. If the need for consent was waived by the ethics committee, please include this information.

“I have read the journal's policy and the authors of this manuscript have the following competing interests: PR works as the scientific coordinator of the German Resuscitation Registry. JW, JTG, MF, AB, SB, and SS are members of the steering committee of the German Resuscitation Registry. JTG is the spokesperson of the steering committee.”

**Additional Editor Comments:**

Thank you for the submission. I apologize for the delay, the paper is of interest. It needs just minor revision.

Reviewers' comments:

Reviewer's Responses to Questions

**Comments to the Author**

1. Is the manuscript technically sound, and do the data support the conclusions?

Reviewer #1: Yes

2. Has the statistical analysis been performed appropriately and rigorously? 

Reviewer #1: Yes

3. Have the authors made all data underlying the findings in their manuscript fully available?

Reviewer #1: Yes

4. Is the manuscript presented in an intelligible fashion and written in standard English?

Reviewer #1: Yes

5. Review Comments to the Author

Reviewer #1: Dear authors , thanks for your paper.

Few critiscism:

-at line 47 is not fully clear the proportion of the OHCA

-it might be interesting know the number of COVId positive patients in the cohorts to understand where COVID is implied in the incidence of the event

-is the high quality number of cases influenced by the pandemic period or the proportion is the same?

6. PLOS authors have the option to publish the peer review history of their article (what does this mean?). If published, this will include your full peer review and any attached files.

Reviewer #1: No

---

## [Author Response · Author response to Decision Letter 0]

13 Jul 2022

Dear Dr. Ballotta,

dear reviewer,

thank you for the time and effort you have put into reviewing our manuscript entitled “Impact of COVID-19 on out-of-hospital cardiac arrest: a registry-based cohort-study from the German Resuscitation Registry”.

We would like to respond to your comments in detail below. You can see all the changes in the attached version with the track changes. 

Comment #1: At line 47 is not fully clear the proportion of the OHCA

Answer #1: Thank you for the comment. We have worded it more clearly.

Comment #2: It might be interesting know the number of COVID positive patients in the cohorts to understand where COVID is implied in the incidence of the event

Answer #2: To our regret, the COVID status of patients is not surveyed by the German Resuscitation Register. Therefore, we are unfortunately unable to make any statements about this.

Comment #3: Is the high quality number of cases influenced by the pandemic period or the proportion is the same?

Answer #3: In the pandemic period, the share of cases from the high quality data group in the total cases was lower than in the non-pandemic period. We took this into account by comparing only those sites that belonged to the high quality data group in both periods.

Furthermore, where necessary, we have adapted formatting according to the specifications of PLOS One.

Our COIs presented in our COI forms do not alter our adherence to PLOS ONE policies on sharing data and materials. We welcome PLOS One's policy of publishing a minimal data set with the article. You will find this attached for publication (S1 table and S2 table).

Please feel free to contact me at any time if you have any queries concerning our manuscript.

Thank you for your consideration. I look forward to hearing from you.

Sincerely,

Stephan Seewald, MD

On behalf of all authors.

University Hospital Schleswig-Holstein

Institute for Emergency Medicine and Department of Anesthesiology and Intensive Care Medicine

Arnold-Heller-Straße 3

24105 Kiel, Germany

Phone +49 431 500-31551

Fax +49 431 500-31504

E-mail stephan.seewald@uksh.de

---

## [Decision Letter · Decision Letter 1]

26 Aug 2022

Impact of COVID-19 on Out-of-Hospital Cardiac Arrest: A Registry-Based Cohort-Study from the German Resuscitation Registry

PONE-D-22-03386R1

Dear Dr. Seewald,

We’re pleased to inform you that your manuscript has been judged scientifically suitable for publication and will be formally accepted for publication once it meets all outstanding technical requirements.

Kind regards,

Andrea Ballotta

Academic Editor

PLOS ONE

Additional Editor Comments (optional):

Tx again for your contribution, the manuscript can be accepted for publication

Reviewers' comments:

Reviewer's Responses to Questions

**Comments to the Author**

1. If the authors have adequately addressed your comments raised in a previous round of review and you feel that this manuscript is now acceptable for publication, you may indicate that here to bypass the “Comments to the Author” section, enter your conflict of interest statement in the “Confidential to Editor” section, and submit your "Accept" recommendation.

Reviewer #1: All comments have been addressed

2. Is the manuscript technically sound, and do the data support the conclusions?

Reviewer #1: Yes

3. Has the statistical analysis been performed appropriately and rigorously? 

Reviewer #1: Yes

4. Have the authors made all data underlying the findings in their manuscript fully available?

Reviewer #1: Yes

5. Is the manuscript presented in an intelligible fashion and written in standard English?

Reviewer #1: Yes

6. Review Comments to the Author

Reviewer #1: Thanks for your specificication and correction, you have address all the criticism ity is a pity you have no idea of the COVID proportrion

7. PLOS authors have the option to publish the peer review history of their article (what does this mean?). If published, this will include your full peer review and any attached files.

Reviewer #1: No

---

## [Editor Report · Acceptance letter]

1 Sep 2022

PONE-D-22-03386R1 

Impact of COVID-19 on out-of-hospital cardiac arrest: a registry-based cohort-study from the German Resuscitation Registry 

Dear Dr. Seewald:

I'm pleased to inform you that your manuscript has been deemed suitable for publication in PLOS ONE. Congratulations! Your manuscript is now with our production department. 

Kind regards, 

on behalf of

Dr. Andrea Ballotta 

Academic Editor

PLOS ONE